# A Characteristic Function for Shapley-Value-Based Attribution of Anomaly Scores

**Naoya Takeishi**  *ntake@g.ecc.u-tokyo.ac.jp*
*The University of Tokyo*
*RIKEN Center for Advanced Intelligence Project*

**Kawahara Yoshinobu**  *kawahara@ist.osaka-u.ac.jp*
*Osaka University*
*RIKEN Center for Advanced Intelligence Project*

**Reviewed on OpenReview:** *https://openreview.net/forum?id=eLX5XrajXh*

## Abstract

In anomaly detection, the degree of irregularity is often summarized as a real-valued anomaly score. We address the problem of attributing such anomaly scores to input features for interpreting the results of anomaly detection. We particularly investigate the use of the Shapley value for attributing anomaly scores of semi-supervised detection methods. We propose a characteristic function specifically designed for attributing anomaly scores. The idea is to approximate the absence of some features by locally minimizing the anomaly score with regard to the to-be-absent features. We examine the applicability of the proposed characteristic function and other general approaches for interpreting anomaly scores on multiple datasets and multiple anomaly detection methods. The results indicate the potential utility of the attribution methods including the proposed one.

## 1 Introduction

Anomaly detection has been one of the major tasks of machine learning and data mining, and a number of different methods have been proposed (see, e.g., Chandola et al., 2009; Pimentel et al., 2014). Anomaly localization (or root cause analysis) is a closely related task whose goal is to identify which features are the most responsible for the irregularity of anomaly-detected data. It is obviously an essential problem in many applications; understanding why specific instances have been considered anomalous is highly valuable for decision-making. Rigorous localization of anomalies is possible only with some physical or causal models of a data-generating process (Budhathoki et al., 2022). However, such models are not always available for complex natural and artificial systems, for which learning-based anomaly detection often has a high demand. Hence, we focus on the latter, that is, the structure-agnostic setting.

We focus on the structure-agnostic interpretation of anomaly detection results, instead of the rigorous anomaly localization based on causal structures. We particularly study methods for post hoc attribution of anomaly scores to input features. In recent years, post hoc attribution of machine learning models, not only in anomaly detection but in general, is an active area of study. One of the most popular approaches refers to the notion of the Shapley value, which has been originally discussed in game theory (Shapley, 1953) and applied to machine learning (Lipovetsky, 2006; Štrumbelj & Kononenko, 2014; Datta et al., 2016; Lundberg & Lee, 2017; Sundararajan et al., 2017; Owen & Prieur, 2017; Ancona et al., 2019; Olsen et al., 2022). It has also been utilized for anomaly interpretation (Antwarg et al., 2021; Giurgiu & Schumann, 2019; Takeishi, 2019). These studies have concluded affirmatively on the use of the Shapley value for the interpretation of anomaly detection based on the attribution of anomaly scores.

The previous studies on using the Shapley value for anomaly score attribution adopted general definitions of the characteristic function of the Shapley value. The characteristic function, namely $v : 2^d \to \mathbb{R}$, is a

set function that represents a game's gain under each coalition of players. In using the Shapley value for attributing machine learning models, it takes a subset of features (and their values) and should return the behavior of the model only with the selected features given as the model's input. Several working definitions of such a characteristic function for general machine learning models have been studied (see, e.g., Lundberg & Lee, 2017; Sundararajan & Najmi, 2020). However, when it comes to applying them to the interpretation of anomaly detection, they do not take into account the specific nature of anomaly scores, which might limit the utility of the attribution method.

In this paper, we propose a definition of the characteristic function for the Shapley value particularly for attributing anomaly scores. The idea is to approximate the absence of some features by minimizing an anomaly score in the proximity of the original data point to be interpreted. As the exact computation of such a characteristic function is usually prohibitively time-consuming, we also present practical relaxed definitions. We empirically examine not only the proposed characteristic function but also some existing definitions of the characteristic function by applying them to anomaly scores. We show the results of experiments on synthetic and real anomalies with multiple datasets and anomaly detection methods to discuss the applicability of attribution methods.

## 2 Background

### 2.1 Anomaly detection

We focus on the type of anomaly detection methods where some *anomaly score* is computed. Semi-supervised anomaly detection[1] usually falls into this category, where a mechanism to detect anomalies is sought given only normal data. A typical solution of semi-supervised anomaly detection consists of two phases (see, e.g., Sections 4, 6, and 7 of Chandola et al., 2009). In the first phase (i.e., the training phase), a model, such as a density estimator, a subspace-based model, and a set of clusters, is learned with the normal data. Then, in the second phase (i.e., the test phase), an anomaly score

$$e : \mathcal{X} \to \mathbb{R}$$

is computed using the learned model, where $\boldsymbol{x} \in \mathcal{X}$ is a test data sample in some data space $\mathcal{X}$ with $d$ features such as $\mathcal{X} \subset \mathbb{R}^d$ or $\mathcal{X} \subset \{0, 1\}^d$. The anomaly score, $e(\boldsymbol{x})$, should indicate a large value when $\boldsymbol{x}$ is anomalous. An alarm is issued if $e(\boldsymbol{x})$ exceeds some threshold value, $e_{\text{th}}$. Semi-supervised anomaly detection methods are useful in many practices, since they do not assume labeled anomalies and generally do not require storing all the data after the training phase.

A popular choice of $e(\boldsymbol{x})$ is the negative log-likelihood, $e(\boldsymbol{x}) = -\log q(\boldsymbol{x})$, where $q$ is some approximation of data's density by models such as Gaussian mixture models (GMMs). Another popular choice is the reconstruction error of dimensionality reduction methods such as principal component analysis (PCA) or autoencoders, $e(\boldsymbol{x}) = \|\boldsymbol{x} - \boldsymbol{g}(\boldsymbol{f}(\boldsymbol{x}))\|^2$, where $\boldsymbol{f}$ and $\boldsymbol{g}$ are an encoder and a decoder between the data and latent spaces, respectively. Moreover, these two views are not mutually exclusive. For example, Zong et al. (2018) reported a good detection performance by using both the energy of the latent representations and the reconstruction errors of an autoencoder.

A straightforward way to interpret such anomaly scores is to examine decomposed or marginal values of the scores. For example, reconstruction errors in Euclidean space are inherently decomposable into the errors of individual features. We may also refer to the marginal likelihood of each feature if the model admits such decomposition. However, we cannot always compute such quantities in general; for example, marginal distributions are usually not tractable for nonlinear generative models.

### 2.2 Attribution with the Shapley value

The Shapley value (Shapley, 1953) is a concept solution of coalitional games. Let $v : 2^d \to \mathbb{R}$ be a set function that represents a game's gain obtained from each coalition (i.e., a subset) of the game's players, and

---

[1]Following the terminology of Chandola et al. (2009); synonymous to novelty detection (e.g., Pimentel et al., 2014).

let $D = \{1, \ldots, d\}$ denote the set of all players. This function $v$ is called a *characteristic function*. A game is defined as a pair $(v, D)$. The Shapley value of $(v, D)$ is to distribute the total gain $v(D)$ to each player in accordance with each one's contribution. The Shapley value of the player $i \in D$, namely $\phi_i$, is the weighted average of the marginal contributions, that is,

$$\phi_i = \sum_{S \subseteq D \setminus \{i\}} \frac{|S|!(d - |S| - 1)!}{d!} \left( v(S \cup \{i\}) - v(S) \right), \tag{1}$$

where $S$ denotes a subset of $D \setminus \{i\} = \{1, ..., i-1, i+1, ..., d\}$, and the sum is taken over all the subsets. The Shapley value has been utilized for interpreting outputs of statistical machine learning (e.g., Lipovetsky, 2006; Štrumbelj & Kononenko, 2014; Datta et al., 2016; Lundberg & Lee, 2017; Sundararajan et al., 2017; Owen & Prieur, 2017; Ancona et al., 2019; Olsen et al., 2022), where the players of a game mean input features, and the gain of the game means the output of a machine learning model.

Major challenges in utilizing the Shapley value include the following two points:

(i) How to compute the summation for $O(2^d)$ terms (i.e., $\sum_{S \subseteq D \setminus \{i\}}$ in Eq. (1))?

(ii) How to define a characteristic function $v$?

The former challenge, the exponential complexity, is a general difficulty and not limited to machine learning interpretation. A common remedy is Monte Carlo approximations (see, e.g., Castro et al., 2017, and references therein). In this work, we also use a Monte Carlo approximation and compute $\{\phi_i\}$ using the reformulation of Eq. (1) as a weighted least squares problem (Charnes et al., 1988; Lundberg & Lee, 2017). We will describe its concrete procedures later in Section 3.2.

The latter challenge, the definition of $v$, rises specifically in interpreting machine learning because $v(S)$ should simulate the "absence" of the features not in $S$ for a machine learning model. It is, in principle, a question that admits no unique solutions; the most straightforward definition would be based on re-training of models for all subsets of $D$, which is not realistic in practice. We will see the existing approaches in the next subsection, Section 2.3.

## 2.3 Approaches to defining characteristic function

In the use of the Shapley value for machine learning model attribution in general, a characteristic function $v$ has often been defined in either of the following two approaches:

**Reference-based approach** replaces the values of "absent" features by some reference values (Sundararajan et al., 2017; Lundberg & Lee, 2017; Ancona et al., 2019). Suppose $\boldsymbol{x} = [x_1, \ldots, x_d] \in \mathbb{R}^d$. Let us denote the subvector of $\boldsymbol{x}$ corresponding to index set $S \subset \{1, \ldots, d\}$ by $\boldsymbol{x}_S \in \mathbb{R}^{|S|}$. Let $S^c := \{1, \ldots, d\} \setminus S$ be the complement of $S$. Then, the value of $v$ for a machine learning model $h : \mathbb{R}^d \to \mathcal{Y}$, where $\mathcal{Y}$ is some output space, is defined as the value of $h$ on a sample where $\boldsymbol{x}_{S^c}$ is replaced by some reference value $\boldsymbol{r}_{S^c} \in \mathbb{R}^{|S|}$. A challenge here is to choose a good reference vector $\boldsymbol{r}_{S^c}$. $\boldsymbol{r}_{S^c}$ is often set to be zero or average of $\boldsymbol{x}_{S^c}$, but there is no unique definition.

**Marginalization-based approach** marginalizes out "absent" features (Štrumbelj & Kononenko, 2014; Datta et al., 2016; Lundberg & Lee, 2017; Olsen et al., 2022). That is, $v$ is computed as the conditional expectation of $h(\boldsymbol{x})$ given $\boldsymbol{x}_S$, where $\boldsymbol{x}_{S^c}$ is marginalized over some distribution $p(\boldsymbol{x}_{S^c} \mid \boldsymbol{x}_S)$. A challenge here is the computation of the conditional expectation, which is intractable in general. Typically, it is approximated via nearest neighbors in training data. Meanwhile, Sundararajan & Najmi (2020) argues that the marginalization-based approach loses some nice properties as attribution because it depends not only on the target function $h$ but also on the data distribution. If the features are independent, this approach reduces to (the average of) the reference-based one with multiple reference vectors.

Apart from the general point of view, we now briefly review how the Shapley value has been used for attributing anomaly scores. Giurgiu & Schumann (2019) and Antwarg et al. (2021) adopted the reference-based approach to defining $v$ for anomaly scores. Since a good reference vector $\boldsymbol{r}.$ depends on a query data point $\boldsymbol{x}$ and a target feature set $S$, it should be determined adaptively to both $\boldsymbol{x}$ and $S$. However, in Antwarg et al. (2021), the reference does not depend on $\boldsymbol{x}$ nor $S$ in principle. Giurgiu & Schumann (2019) proposed to

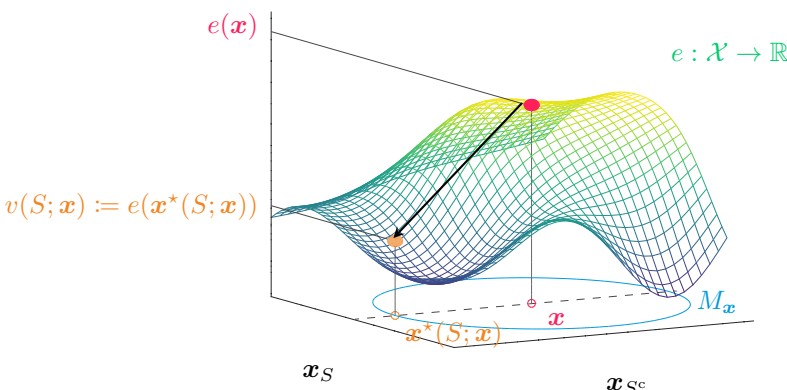

Figure 1: Proposed characteristic function, $v(S; \boldsymbol{x})$ in Eq. (2), for attributing an anomaly score $e : \mathcal{X} \to \mathbb{R}$. $\boldsymbol{x}$ has $d$ features, which are split into disjoint sets $S \subset \{1, \ldots, d\}$ and $S^{\mathrm{c}} = \{1, \ldots, d\} \backslash S$. $v(S; \boldsymbol{x})$ is defined as a local minimum of $e(\boldsymbol{x})$ in the proximity of $\boldsymbol{x}$ (i.e., $M_{\boldsymbol{x}}$) with $\boldsymbol{x}_S$ fixed at the original. Although the axes of $\boldsymbol{x}_S$ and $\boldsymbol{x}_{S^{\mathrm{c}}}$ are depicted as one dimension, they are $|S|$- and $|S^{\mathrm{c}}|$-dimensional in general.

choose a reference value adaptively using the influence weights between a query data point $\boldsymbol{x}$ and a training dataset. Such a reference is adaptive to $\boldsymbol{x}$ but not to $S$, meaning the same reference value is used for every $S$. We also note that their method requires storing enough portion of training data, which may be undesirable in some applications of semi-supervised anomaly detection.

Takeishi (2019) adopted the marginalization-based approach to defining $v$ for the anomaly score computed with the probabilistic principal component analysis. Because the conditional expectation is available for the probabilistic PCA models, the marginalization can be computed exactly. This approach is free from choosing a reference value, but the type of applicable anomaly detection models is obviously restrictive. Moreover, considering conditional distributions under the presence of anomalies is not necessarily meaningful because the learned distribution is no longer reliable for out-of-distribution data points.

## 3 Anomaly score attribution with the Shapley value

We propose a definition of the characteristic function that takes into account the nature of anomaly scores. We take the reference-based approach; differently from the existing studies, the proposed method determines the reference value, $\boldsymbol{r}_{.}$, adaptively to both $\boldsymbol{x}$ and $S$ and without referring to training data.

### 3.1 A characteristic function for anomaly scores

Let $e : \mathcal{X} \to \mathbb{R}$ be an anomaly score function that outputs a large value when the input, $\boldsymbol{x} \in \mathcal{X} \subset \mathbb{R}^d$, is anomalous. We propose to define a characteristic function, which we will denote by $v(S; \boldsymbol{x})$ to manifest the dependency on $\boldsymbol{x}$, for an anomaly score $e$ as follows. Recall that we want to design $v(S; \boldsymbol{x})$ that simulates the "absence" of the features not in $S$. In other words, $v(S; \boldsymbol{x})$ should represent how anomalous $\boldsymbol{x}_S$ is with the remaining features $\boldsymbol{x}_{S^{\mathrm{c}}}$ ignored. To this end, we define $v(S; \boldsymbol{x})$ as *the smallest value of the anomaly score, e, achieved in a neighborhood of $\boldsymbol{x}$ with $\boldsymbol{x}_S$ being fixed.* Rephrasing the idea, $v(S; \boldsymbol{x})$ is defined as follows:

$$v(S; \boldsymbol{x}) \coloneqq e(\boldsymbol{x}^{\star}(S; \boldsymbol{x})), \quad \text{where} \quad \boldsymbol{x}^{\star}(S; \boldsymbol{x}) \coloneqq \arg\min_{\boldsymbol{y}} e(\boldsymbol{y}) \quad \text{s.t.} \quad \boldsymbol{y} \in M_{\boldsymbol{x}} \subset \mathcal{X}, \boldsymbol{y}_S = \boldsymbol{x}_S. \quad (2)$$

The constraint of the optimization, $M_{\boldsymbol{x}} \subset \mathcal{X}$, is some compact neighborhood of $\boldsymbol{x} \in \mathcal{X}$. Figure 1 illustrates the idea; while $\boldsymbol{x}_S$ is fixed at the original value, $\boldsymbol{x}_{S^{\mathrm{c}}}$ is moved within $M_{\boldsymbol{x}}$ so that the value of $e$ is minimized.

The characteristic function in Eq. (2) enables us to examine how $e(\boldsymbol{x})$ becomes large solely due to the features in $S$. Although it falls in the category of reference-based characteristic functions, it is different from the general methods in which the absence of features is simulated by the replacement with *predefined* reference

values (Antwarg et al., 2021; Giurgiu & Schumann, 2019). The proposed method automatically determines the reference values depending on $\boldsymbol{x}_S$ via solving the optimization problem in Eq. (2).

Practically, it is unrealistic to determine $M_{\boldsymbol{x}}$ manually because it should depend both on the geometry of $\mathcal{X}$ and on the property of $e(\boldsymbol{x})$. Hence, we propose a variant of Eq. (2) that admits less complexity of manual tuning. Instead of taking a minimum in a compact neighborhood, we consider a local minimum of $e$ regularized by the distance from the original $\boldsymbol{x}_S$. That is, we define a variant of $v$, namely $\hat{v}$, as follows:

$$\hat{v}(S; \boldsymbol{x}) \coloneqq e(\hat{\boldsymbol{x}}^{\star}(S; \boldsymbol{x})), \quad \text{where} \quad \hat{\boldsymbol{x}}^{\star}(S; \boldsymbol{x}) \coloneqq \arg\min_{\boldsymbol{y}} \ell_{S, \boldsymbol{x}}(\boldsymbol{y}) \quad \text{s.t.} \quad \boldsymbol{y} \in \mathcal{X}, \, \boldsymbol{y}_S = \boldsymbol{x}_S,$$
$$\text{and} \quad \ell_{S, \boldsymbol{x}}(\boldsymbol{y}) \coloneqq e(\boldsymbol{y}) + \frac{\gamma}{|S^{\mathrm{c}}|} \sum_{i \in S^{\mathrm{c}}} \mathrm{dist}(y_i, x_i). \tag{3}$$

The second term of $\ell_{S, \boldsymbol{x}}$ is a regularizer. The hyperparameter, $\gamma \geq 0$, controls how far $\boldsymbol{x}_{S^{\mathrm{c}}}$ can move from the original value and semantically corresponds to the radius of $M_{\boldsymbol{x}}$ in Eq. (2). Selecting the value of $\gamma$ is not straightforward in general because it depends on the geometry of the data space. A practical suggestion is to roughly set a reference value of $\gamma$ such that the magnitude of the two terms of $\ell_{S, \boldsymbol{x}}$ become similar and try different values around it. To evaluate each $\gamma$, one can create anomalous data artificially based on prior knowledge and see how well the artificial anomalies are attributed. Meanwhile, we empirically found that the performance of the attribution method was not so sensitive to the value of $\gamma$, as reported in Appendix B.1. $\mathrm{dist}(\cdot, \cdot)$ is some dissimilarity function in each dimension of $\mathcal{X}$. It is to be defined by a user in accordance with the nature of the data space, $\mathcal{X}$. We set it as the Euclidean distance in the experiments in Section 5. Comparing Eq. (2) and Eq. (3), we can understand that the constraint in Eq. (2) (i.e., $\boldsymbol{y} \in M_{\boldsymbol{x}}$) is relaxed to be the regularizer in Eq. (3) (i.e., the second term of $\ell_{S, \boldsymbol{x}}$).

Despite the relaxation from $v$ in Eq. (2) to $\hat{v}$ in Eq. (3), the computation can still be prohibitively heavy for a moderate number of features (e.g., $d \gtrsim 10$) because computing $\hat{v}$ needs to run the local minimization for $O(2^d)$ times. We suggest a heuristic relaxation that reduces the number of minimization runs to $O(d)$. The bottleneck lies in computing $\hat{\boldsymbol{x}}^{\star}(S; \boldsymbol{x})$, a local minimum of $\ell_{S, \boldsymbol{x}}$. Hence, instead of $\hat{\boldsymbol{x}}^{\star}$, we define an "ansatz" $\underline{\boldsymbol{x}}^{\star}$ as follows:

$$\underline{\boldsymbol{x}}^{\star}(S; \boldsymbol{x}) \coloneqq \boldsymbol{z} \quad \text{s.t.} \quad \boldsymbol{z}_S = \boldsymbol{x}_S \quad \text{and} \quad \boldsymbol{z}_{S^{\mathrm{c}}} = \frac{1}{|S| + 1} \left( \hat{\boldsymbol{x}}^{\star}(\varnothing; \boldsymbol{x}) + \sum_{i \in S} \hat{\boldsymbol{x}}^{\star}(\{i\}; \boldsymbol{x}) \right). \tag{4}$$

The computation process in Eq. (4) can be rephrased in words as follows. The subvector of $\underline{\boldsymbol{x}}^{\star}(S; \boldsymbol{x})$ corresponding to $S$ (which we will call the $S$-subvector of $\underline{\boldsymbol{x}}^{\star}$) is from the original $\boldsymbol{x}$ (i.e., $\boldsymbol{x}_S$). This is the same with the $S$-subvector of $\hat{\boldsymbol{x}}^{\star}$, and the difference between $\hat{\boldsymbol{x}}^{\star}$ and $\underline{\boldsymbol{x}}^{\star}$ lies in the remaining part, the $S^{\mathrm{c}}$-subvector. While the $S^{\mathrm{c}}$-subvector of $\hat{\boldsymbol{x}}^{\star}$ is defined directly as a minimizer of $\ell_{S, \boldsymbol{x}}$, the $S^{\mathrm{c}}$-subvector of $\underline{\boldsymbol{x}}^{\star}$ is defined as the average of $\hat{\boldsymbol{x}}^{\star}$ computed for the singletons of $S$'s elements and the empty set. It means that we need to compute the minimization in Eq. (3) only for $|S| + 1$ times and that we can reuse its results for every $S$ afterward to compute $\underline{\boldsymbol{x}}^{\star}$. Consequently, we define a relaxed characteristic function, namely $\underline{v}$, as

$$\underline{v}(S; \boldsymbol{x}) \coloneqq e(\underline{\boldsymbol{x}}^{\star}(S; \boldsymbol{x})). \tag{5}$$

The relaxation from $\hat{\boldsymbol{x}}^{\star}$ in Eq. (3) to $\underline{\boldsymbol{x}}^{\star}$ in Eq. (4) is heuristic, and the latter is not necessarily an approximation of the former in a proper sense. Meanwhile, we empirically found that the heuristic relaxation still works to some extent, as reported in Section 5. We note the idea of the relaxation as follows. First, recall that the original definition in Eq. (3) is the solution of the minimization problem with *all* the elements of $\boldsymbol{x}$ indexed in $S$ fixed. Then, in Eq. (4), we try to make its surrogate by averaging the local solution of minimization problems with *each* element of $\boldsymbol{x}$ indexed in $S$ fixed in each problem. We also counted the empty set as one of the cases of this process, thus the $|S| + 1$ cases in Eq. (4) appeared.

## 3.2 Overall algorithm

The main contribution of this paper lies in the definition of the characteristic function for attributing anomaly scores, which we have presented so far. Meanwhile, we note the overall algorithm to approximately compute

---

**Algorithm 1** Shapley value approximation for anomaly score attribution

---

**Input:** Anomaly score function $e$, query data point $\boldsymbol{x}$, hyperparameter $\gamma \geq 0$, number of MC samples $m$
**Output:** Shapley values for each feature, $\{\phi_1, \ldots, \phi_d\}$

1: **for** $s = \varnothing, \{1\}, \ldots, \{d\}$ **do**
2: $\quad \hat{\boldsymbol{x}}^{\star}(s; \boldsymbol{x}) \leftarrow \underset{\boldsymbol{y} \in \mathcal{X}, \, \boldsymbol{y}_s = \boldsymbol{x}_s}{\arg\min} \ \ell_{s, \boldsymbol{x}}(\boldsymbol{y})$ $\qquad\qquad\qquad\qquad\qquad\qquad \triangleright \ell_{S, \boldsymbol{x}}$ in Eq. (3)
3: **end for**
4: **for** $j = 1, \ldots, m$ **do**
5: $\quad S_j \leftarrow$ Draw a random subset $S$ of $\{1, \ldots, d\}$ with probability $\propto (d-1)|S|!(d - |S| - 1)!/d!$
6: $\quad \underline{\boldsymbol{x}}^{\star}(S_j; \boldsymbol{x}) \leftarrow$ Eq. (4)
7: **end for**
8: $\{\phi\} \leftarrow \arg\min_{\phi'} \sum_j \left( e(\underline{\boldsymbol{x}}^{\star}(S_j; \boldsymbol{x})) - \left( \sum_{i \in \{0\} \cup S_j} \phi'_i \right) \right)^2$ $\quad \triangleright$ Charnes et al. (1988); Lundberg & Lee (2017)

---

the Shapley value, in Algorithm 1, for the completeness of the paper. The following are some notes on the algorithm:

- It does not assume any specific types of anomaly detectors as long as a (sub)differentiable anomaly score function is defined. The model can be GMMs, autoencoders, their combinations or ensembles, or anything else, and the anomaly score can be negative log-likelihoods, reconstruction errors, or others.

- Lines 4–8 of Algorithm 1 perform the Monte Carlo approximation of the Shapley value computation. Upon the approximation, we utilize the weighted least squares formulation of the Shapley value (Charnes et al., 1988; Lundberg & Lee, 2017; Aas et al., 2021).

- The only hyperparameter specific to the proposed method is $\gamma$. We found that the performance was not sensitive to $\gamma$ (see the appendix). $m$ is a hyperparameter generally present in Shapley value approximation, and we used the value recommended in Lundberg & Lee (2017), $m = 2d + 2^{11}$.

## 4 Related Work

As mentioned earlier, anomaly score attribution with the Shapley value has already been studied (Antwarg et al., 2021; Giurgiu & Schumann, 2019; Takeishi, 2019), though the characteristic functions used in these studies are not necessarily specialized for anomaly scores. We will also try general methods of the Shapley value-based attribution in our experiments in Section 5 for comparison.

For semi-supervised anomaly detection, other types of interpretation methods, not explicitly based on the Shapley value, have also been proposed. Siddiqui et al. (2019) formulated the sequential feature explanation, in which they seek a most convincing order of features to explain an anomaly. Zhang et al. (2019) suggested using a linear surrogate model learned via perturbation around a data point for explaining why the data point was regarded anomalous. We will also examine these methods in Section 5.

The anomaly attribution method of Idé et al. (2021) is notable for its conceptual similarity with ours. Their method is for attributing anomalies found in a regression model $\boldsymbol{x} \mapsto y$, where $y$ is the regressed variable. It seeks a local correction $\boldsymbol{\delta}$ of $\boldsymbol{x}$ such that $\boldsymbol{x} + \boldsymbol{\delta}$ maximizes the likelihood $p(y \mid \boldsymbol{x} + \boldsymbol{\delta})$ under the regression model, and the elements of $\boldsymbol{\delta}$ are regarded as attributions. While the targeted type of anomaly detection is different from ours, their idea reminds us of the intermediate quantity of our approach, $\boldsymbol{x}^{\star}$, in Eq. (2); we seek $\boldsymbol{x}^{\star}$ such that it minimizes the anomaly score in the vicinity of the original input. The method of Idé et al. (2021) could be roughly understood as using $\boldsymbol{\delta} = \boldsymbol{x}^{\star}(\varnothing; \boldsymbol{x}) - \boldsymbol{x}$ as attribution. In contrast, we compute the attribution based on the values of $\boldsymbol{x}^{\star}(S; \boldsymbol{x})$ not only for $S = \varnothing$ but also for different configurations of $S$.

Apart from the semi-supervised setting, many studies have been done on the interpretation of unsupervised anomaly detection (e.g., Knorr & Ng, 1999; Kriegel et al., 2012; Keller et al., 2012; Dang et al., 2016; Vinh et al., 2015; Liu et al., 2018; Yepmo et al., 2022) or *outlier detection*. These methods could be adjusted for the semi-supervised setting, but how it should be done depends on each method. We did not include these methods in the comparison in Section 5, since there already are several methods (as exemplified above)

Table 1: Datasets properties. $d$ is the number of features. Note that $|\mathcal{D}_{\text{test}}| = 2|\mathcal{D}_{\text{test}}^{\text{norm}}| = 2|\mathcal{D}_{\text{test}}^{\text{anom}}|$.

| Name | $d$ | Type | $|\mathcal{D}_{\text{train}}|$ | $|\mathcal{D}_{\text{valid}}|$ | $|\mathcal{D}_{\text{test}}|$ |
|---|---|---|---|---|---|
| Thyroid | 6 | real | 2869 | 717 | 186 |
| BreastW | 9 | real | 164 | 41 | 478 |
| U2R | 10 | real | 12310 | 3078 | 420 |
| Lympho | 59 | binary | 109 | 27 | 12 |
| Musk | 166 | real | 2294 | 574 | 194 |
| Arrhythmia | 274 | real | 256 | 54 | 132 |

that directly fit the semi-supervised setting. Adapting the interpretation methods for outlier detection to semi-supervised anomaly detection would be an independent study.

Budhathoki et al. (2022) address a related problem with a fundamentally different assumption; they assume to have the knowledge of the causal structure behind data. They compute the Shapley value of anomaly scores by referring to a functional causal model. While their method is not applicable to our setting as we assume rather the absence of causal knowledge, exploring new problem settings between the two extrema (complete presence or absence of causal knowledge) would be an interesting future direction.

## 5 Experiments

We present the empirical performance of the proposed method and baseline methods for different datasets and anomaly detectors. We will summarize the implications of the results later in Section 6.

### 5.1 Experimental setting

#### 5.1.1 Datasets

We used six public datasets with different numbers of features as listed in Table 1. The U2R dataset is a subset of the NSL-KDD dataset (Tavallaee et al., 2009), which is a modified version of the KDDCup'99 data. We created the U2R dataset by extracting the U2R attack type from the NSL-KDD dataset and by eliminating categorical and constant-valued features. The other datasets are from the ODDS repository (Rayana, 2016). For the Lympho dataset, we converted the categorical features into binary by one-hot encoding, which resulted in the 59-dimensional dataset.

We split each dataset into a training set $\mathcal{D}_{\text{train}}$, a validation set $\mathcal{D}_{\text{valid}}$, and a test set $\mathcal{D}_{\text{test}} = \mathcal{D}_{\text{test}}^{\text{norm}} \cup \mathcal{D}_{\text{test}}^{\text{anom}}$. We used the whole anomaly part of each dataset as $\mathcal{D}_{\text{test}}^{\text{anom}}$ and randomly chose the same size of normal data as $\mathcal{D}_{\text{test}}^{\text{norm}}$. $\mathcal{D}_{\text{train}}$ and $\mathcal{D}_{\text{valid}}$ were created from the remaining normal data. We used $\mathcal{D}_{\text{train}}$ and $\mathcal{D}_{\text{valid}}$ for training and selection of anomaly detectors, respectively, whereas $\mathcal{D}_{\text{test}}^{\text{norm}}$ and $\mathcal{D}_{\text{test}}^{\text{anom}}$ were secured for measuring the performance of attribution methods. We normalized the real-valued datasets using each training set's mean and standard average.

#### 5.1.2 Anomaly scores

We examined the following types of anomaly detectors with corresponding $e(\boldsymbol{x})$ (details are in the appendix):

**GMM** $\quad e(\boldsymbol{x})$ as the energy (i.e., negative log-likelihood) of GMM.

**VAE-r** $\quad e(\boldsymbol{x})$ as the reconstruction error of VAE.

**VAE-e** $\quad e(\boldsymbol{x})$ as the negative empirical lower bound of VAE.

**DAGMM** $\quad e(\boldsymbol{x})$ as the energy of the deep autoencoding Gaussian mixture models (Zong et al., 2018).

#### 5.1.3 Attribution methods

We tried the following three baseline methods:

**MARG**     Marginal scores of individual features. We examined the energy of marginal distributions for *GMM* and the reconstruction error of each feature for *VAE-r*. In principle, this baseline is not applicable to *VAE-e* and *DAGMM* as they do not admit straightforward decomposition of the anomaly scores.

**SFE**     The sequential feature explanation (Siddiqui et al., 2019). We used the greedy algorithm named "sequential marginal". This method is applicable only to *GMM* and *VAE-r*, as with MARG.

**ACE**     The anomaly contribution explainer (Zhang et al., 2019).

We also tried the following variants of the general Shapley value-based methods:

**IG**     The integrated gradients (Sundararajan et al., 2017), which can be regarded as a realization of the Aumann–Shapley value. We used the mean of training data as a reference value.

**KSH**     The kernel SHAP (Lundberg & Lee, 2017). As suggested by the authors' implementation, we used the cluster centers computed by $k$-means ($k = 8$) on training data as reference values. We note that Antwarg et al. (2021) also used the kernel SHAP for anomaly scores.

**wKSH**     A variant of the kernel SHAP, namely weighted kernel SHAP, in which we selected reference values by $\boldsymbol{x}$'s $k$-nearest neighbors from training data with $k = 8$. This approach is similar to the method proposed by Giurgiu & Schumann (2019), where they used the influence weights instead.

Finally, with regard to our proposed method, we tried two variants:

**COMP**     Attribution as the absolute value of each element of $\boldsymbol{\delta} = \hat{\boldsymbol{x}}^\star(\varnothing; \boldsymbol{x}) - \boldsymbol{x}$, where $\hat{\boldsymbol{x}}^\star$ is the intermediate quantity of the proposed method in Eq. (3). This is conceptually similar to the method of Idé et al. (2021) (anomaly compensation).

**ASH**     The full proposed method, namely anomaly Shapley, as outlined in Algorithm 1. We used the most relaxed definition of our characteristic function, $\underline{v}$ in Eq. (5), and set $\gamma = 0.01$, unless otherwise stated.

## 5.2   Attribution for synthetic anomaly

We investigated the performance of the attribution methods under controlled conditions using the normal half of test data, $\mathcal{D}_{\text{test}}^{\text{norm}}$. We generated synthetic anomalies by perturbing some features of the normal data and then tried to localize them with attribution results. We synthesized the anomalies as follows: for every data point of $\mathcal{D}_{\text{test}}^{\text{norm}}$, we selected a set of $d_{\text{anom}}$ features, and perturbed the values of the selected features by adding noise drawn uniformly from $[-2, -1] \cup [1, 2]$ (for real-valued datasets). We set this magnitude of the noise because the real-valued datasets were normalized to have the unit standard deviation. For the binary-valued dataset (i.e., `Lympho`), instead of adding the real-valued noise, we flipped the values of the selected features.

All the anomaly detectors resulted in similarly good detection performances probably because the synthesized anomalies were obvious, so we focus only on the attribution performance. We defer the results with $d_{\text{anom}} > 1$ to the appendix, though the overall tendency is the same as what we observed from the results with $d_{\text{anom}} = 1$.

For $d_{\text{anom}} = 1$, we report the mean reciprocal rank (MRR) of the attribution of the perturbed feature. This is the average of the reciprocal of the rank of the perturbed feature's attribution in the descending order, so, if the largest value is attributed to the perturbed feature for every data point, then the MRR becomes 1. The MRR values are detailed for all the datasets and the detectors in Table 2. The proposed approach, ASH, is better than or comparable with the other methods in many dataset-detector pairs. It is worth noting that the simple strategy, MARG, works to some extent in most cases when it is applicable, though an obvious drawback is the limited applicability. We report the result with another criterion, Hits@3 (i.e., the proportion of samples where the perturbed feature was in the top-3 attribution), in Table 3 for completeness, where we observe the same tendency.

Table 2: MRR (the larger, the better) of the synthetic anomalies, for $d_{\text{anom}} = 1$. The percentiles shown at the bottom of the table are statistics of the values of all the dataset-detector pairs. We show the percentiles for MARG and SFE just for information and do not compare them with the other methods' statistics because the applicable detectors are different. The same applies to the following tables.

| Setting | | MRR | | | | | | | |
| Dataset | Detector | MARG | SFE | ACE | IG | KSH | wKSH | COMP | ASH |
|---|---|---|---|---|---|---|---|---|---|
| Thyroid | *GMM* | 0.57 | 0.59 | 0.73 | 0.70 | 0.48 | 0.71 | 0.48 | 0.78 |
| | *VAE-r* | 0.75 | 0.75 | 0.77 | 0.75 | 0.58 | 0.77 | 0.75 | 0.75 |
| | *VAE-e* | — | — | 0.74 | 0.76 | 0.56 | 0.77 | 0.46 | 0.75 |
| | *DAGMM* | — | — | 0.39 | 0.23 | 0.38 | 0.46 | 0.66 | 0.51 |
| Breastw | *GMM* | 0.78 | 0.79 | 0.77 | 0.78 | 0.42 | 0.76 | 0.47 | 0.78 |
| | *VAE-r* | 0.67 | 0.67 | 0.71 | 0.83 | 0.66 | 0.81 | 0.77 | 0.77 |
| | *VAE-e* | — | — | 0.72 | 0.86 | 0.66 | 0.80 | 0.52 | 0.85 |
| | *DAGMM* | — | — | 0.37 | 0.13 | 0.38 | 0.68 | 0.49 | 0.57 |
| U2R | *GMM* | 0.84 | 0.82 | 0.70 | 0.86 | 0.36 | 0.90 | 0.57 | 0.86 |
| | *VAE-r* | 0.84 | 0.84 | 0.88 | 0.87 | 0.36 | 0.91 | 0.81 | 0.89 |
| | *VAE-e* | — | — | 0.83 | 0.89 | 0.37 | 0.90 | 0.41 | 0.88 |
| | *DAGMM* | — | — | 0.58 | 0.11 | 0.29 | 0.74 | 0.78 | 0.71 |
| Lympho | *GMM* | 0.48 | 0.56 | 0.16 | 0.89 | 0.08 | 0.08 | 0.15 | 0.73 |
| | *VAE-r* | 0.92 | 0.92 | 0.16 | 0.66 | 0.08 | 0.08 | 0.03 | 0.87 |
| | *VAE-e* | — | — | 0.16 | 0.69 | 0.08 | 0.08 | 0.03 | 0.83 |
| | *DAGMM* | — | — | 0.08 | 0.12 | 0.08 | 0.08 | 0.16 | 0.37 |
| Musk | *GMM* | 0.08 | 0.10 | 0.24 | 0.28 | 0.20 | 0.92 | 0.11 | 0.97 |
| | *VAE-r* | 0.98 | 0.98 | 0.60 | 0.80 | 0.54 | 0.98 | 0.20 | 0.98 |
| | *VAE-e* | — | — | 0.61 | 0.80 | 0.54 | 0.98 | 0.11 | 0.97 |
| | *DAGMM* | — | — | 0.10 | 0.03 | 0.06 | 0.47 | 0.11 | 0.28 |
| Arrhythmia | *GMM* | 0.22 | 0.24 | 0.25 | 0.42 | 0.09 | 0.48 | 0.08 | 0.49 |
| | *VAE-r* | 0.72 | 0.72 | 0.35 | 0.54 | 0.36 | 0.58 | 0.17 | 0.71 |
| | *VAE-e* | — | — | 0.35 | 0.54 | 0.36 | 0.58 | 0.18 | 0.72 |
| | *DAGMM* | — | — | 0.13 | 0.00 | 0.06 | 0.22 | 0.08 | 0.17 |
| 25th percentile | | (0.55) | (0.58) | 0.22 | 0.27 | 0.09 | 0.47 | 0.11 | **0.67** |
| 50th percentile | | (0.73) | (0.73) | 0.48 | 0.69 | 0.36 | 0.72 | 0.30 | **0.76** |
| 75th percentile | | (0.84) | (0.82) | 0.72 | 0.81 | 0.49 | 0.83 | 0.53 | **0.86** |

## 5.3 Comparison to attribution of supervised classifier

For assessing the performance of the attribution methods in a more realistic situation, we use another half of the test data, $\mathcal{D}_{\text{test}}^{\text{anom}}$. Although we know that $\mathcal{D}_{\text{test}}^{\text{anom}}$ comprises somewhat anomalous data points, we do not know which features are most anomalous for each data point, so the ground truth of anomaly score attribution is not available. Hence, we resort to comparing the attributions of the semi-supervised anomaly scores with the attribution of a supervised classifier's outputs, expecting that the supervised model can capture richer information on the data than the semi-supervised models can do, since the former is explicitly informed by both normal and anomalous labeled data, while the latter is only informed by normal data.

For each dataset, we first train a binary classifier (specifically, SVM with the RBF kernel) using all the data, with the two classes being normal ($\mathcal{D}_{\text{train}} \cup \mathcal{D}_{\text{valid}} \cup \mathcal{D}_{\text{test}}^{\text{norm}}$) and anomalous ($\mathcal{D}_{\text{test}}^{\text{anom}}$).[2] As the two classes are imbalanced, we preprocessed the data using SMOTE (Chawla et al., 2002). We then compute the attribution of the classifier's output for each data point of $\mathcal{D}_{\text{test}}^{\text{anom}}$ using kernel SHAP (Lundberg & Lee, 2017) and choose the feature with the largest absolute value of the attribution as the most anomalous feature for each data point. We only use instances for which the following holds:

$$|\phi_{i_1}^{\text{sup}}| \geq 2|\phi_{i_2}^{\text{sup}}|, \tag{6}$$

---

[2] We use $\mathcal{D}_{\text{test}}^{\text{anom}}$ both for training and attribution, which is not problematic because our purpose does not lie in predicting the label by the supervised classifier; it just works as a reference of attribution.

Table 3: Hits@3 (the larger, the better) of the synthetic anomalies, for $d_{\mathrm{anom}} = 1$.

| Setting | | Hits@3 | | | | | | | |
|---------|----------|------|-----|------|------|------|------|------|------|
| Dataset | Detector | MARG | SFE | ACE | IG | KSH | wKSH | COMP | ASH |
| Thyroid | GMM | 0.79 | 0.81 | 0.85 | 0.89 | 0.66 | 0.89 | 0.46 | 0.88 |
| | VAE-r | 0.86 | 0.86 | 0.88 | 0.86 | 0.78 | 0.84 | 0.86 | 0.86 |
| | VAE-e | — | — | 0.85 | 0.85 | 0.72 | 0.85 | 0.44 | 0.85 |
| | DAGMM | — | — | 0.39 | 0.15 | 0.48 | 0.47 | 0.79 | 0.57 |
| Breastw | GMM | 0.85 | 0.90 | 0.85 | 0.88 | 0.55 | 0.82 | 0.44 | 0.88 |
| | VAE-r | 0.78 | 0.78 | 0.76 | 0.92 | 0.71 | 0.89 | 0.89 | 0.80 |
| | VAE-e | — | — | 0.78 | 0.95 | 0.72 | 0.88 | 0.51 | 0.93 |
| | DAGMM | — | — | 0.32 | 0.03 | 0.37 | 0.69 | 0.53 | 0.68 |
| U2R | GMM | 0.97 | 0.93 | 0.74 | 0.97 | 0.47 | 0.91 | 0.57 | 0.97 |
| | VAE-r | 0.97 | 0.97 | 0.94 | 0.96 | 0.47 | 0.90 | 0.97 | 0.96 |
| | VAE-e | — | — | 0.95 | 0.96 | 0.49 | 0.90 | 0.41 | 0.96 |
| | DAGMM | — | — | 0.60 | 0.00 | 0.28 | 0.74 | 0.93 | 0.83 |
| Lympho | GMM | 0.41 | 0.63 | 0.14 | 0.98 | 0.06 | 0.06 | 0.14 | 0.98 |
| | VAE-r | 0.95 | 0.95 | 0.14 | 0.73 | 0.06 | 0.06 | 0.00 | 0.96 |
| | VAE-e | — | — | 0.14 | 0.82 | 0.06 | 0.06 | 0.00 | 0.84 |
| | DAGMM | — | — | 0.06 | 0.13 | 0.06 | 0.06 | 0.14 | 0.46 |
| Musk | GMM | 0.05 | 0.07 | 0.24 | 0.28 | 0.25 | 0.93 | 0.14 | 0.97 |
| | VAE-r | 0.98 | 0.98 | 0.64 | 0.82 | 0.60 | 0.99 | 0.24 | 0.99 |
| | VAE-e | — | — | 0.63 | 0.82 | 0.59 | 0.99 | 0.14 | 0.97 |
| | DAGMM | — | — | 0.10 | 0.03 | 0.04 | 0.56 | 0.14 | 0.30 |
| Arrhythmia | GMM | 0.21 | 0.23 | 0.27 | 0.49 | 0.07 | 0.57 | 0.07 | 0.53 |
| | VAE-r | 0.75 | 0.75 | 0.41 | 0.70 | 0.45 | 0.71 | 0.18 | 0.78 |
| | VAE-e | — | — | 0.41 | 0.70 | 0.45 | 0.71 | 0.22 | 0.79 |
| | DAGMM | — | — | 0.14 | 0.00 | 0.04 | 0.26 | 0.06 | 0.17 |
| 25th percentile | | (0.67) | (0.72) | 0.21 | 0.25 | 0.07 | 0.54 | 0.14 | **0.76** |
| 50th percentile | | (0.82) | (0.83) | 0.51 | 0.82 | 0.46 | 0.78 | 0.32 | **0.85** |
| 75th percentile | | (0.95) | (0.94) | 0.80 | 0.90 | 0.59 | 0.89 | 0.54 | **0.96** |

Table 4: MRR of the most-attributed features by the supervised classifier for real anomalies.

| Setting | | MRR | | | | | | | |
|---------|----------|------|-----|------|------|------|------|------|------|
| Dataset | Detector | MARG | SFE | ACE | IG | KSH | wKSH | COMP | ASH |
| Thyroid | GMM | 0.52 | 0.44 | 0.64 | 0.57 | 0.54 | 0.75 | 0.68 | 0.91 |
| | VAE-r | 0.87 | 0.87 | 0.86 | 0.87 | 0.71 | 0.76 | 0.87 | 0.87 |
| | VAE-e | — | — | 0.82 | 0.87 | 0.77 | 0.75 | 0.68 | 0.87 |
| | DAGMM | — | — | 0.47 | 0.18 | 0.42 | 0.36 | 0.64 | 0.68 |
| U2R | GMM | 0.72 | 0.69 | 0.48 | 0.73 | 0.37 | 0.45 | 0.82 | 0.73 |
| | VAE-r | 0.64 | 0.64 | 0.70 | 0.65 | 0.38 | 0.37 | 0.71 | 0.66 |
| | VAE-e | — | — | 0.69 | 0.69 | 0.39 | 0.40 | 0.14 | 0.66 |
| | DAGMM | — | — | 0.55 | 0.13 | 0.54 | 0.43 | 0.70 | 0.51 |
| Musk | GMM | 0.02 | 0.03 | 0.26 | 0.01 | 0.05 | 0.16 | 0.64 | 0.04 |
| | VAE-r | 0.06 | 0.06 | 0.24 | 0.14 | 0.20 | 0.12 | 0.63 | 0.10 |
| | VAE-e | — | — | 0.18 | 0.14 | 0.20 | 0.12 | 0.64 | 0.01 |
| | DAGMM | — | — | 0.24 | 0.01 | 0.02 | 0.18 | 0.64 | 0.14 |
| 25th percentile | | (0.17) | (0.15) | 0.26 | 0.14 | 0.20 | 0.17 | **0.64** | 0.13 |
| 50th percentile | | (0.58) | (0.54) | 0.52 | 0.38 | 0.39 | 0.39 | **0.66** | **0.66** |
| 75th percentile | | (0.70) | (0.68) | 0.69 | 0.70 | 0.54 | 0.53 | 0.70 | **0.77** |

where $\phi_{i_1}^{\mathrm{sup}}$ and $\phi_{i_2}^{\mathrm{sup}}$ respectively denote the largest and the second-largest elements of a set $\{\phi_1^{\mathrm{sup}}, \dots, \phi_d^{\mathrm{sup}}\}$, which is the set of the supervised classifier's attribution to the $d$ features. With Eq. (6), we can only use instances for which the $i_1$-th feature is significantly more attributed than the runner-up $i_2$-th feature. We

finally evaluate how well such features can be localized by the attributions of the semi-supervised anomaly scores by methods listed in Section 5.1.3.

In Table 4, we report the MRR of the attribution of the semi-supervised anomaly scores, with the features most attributed in the supervised classifier being the golden standard. The reciprocal rank is 1 when the attribution of the supervised classifier and that of the semi-supervised anomaly score assign the largest value to the same feature. Although ASH is competitive or better compared to other methods for the `Thyroid` and `U2R` datasets, it (as well as the other methods except COMP) fails for the `Musk` dataset. Meanwhile, the COMP baseline method is still successful to some extent for the `Musk` dataset. This is interesting, but the current data resource does not allow further analysis of why this was the case. We report the results only for the three datasets because, for the remaining datasets, only a few instances of each of them sufficed the condition in Eq. (6), and thus the performance comparison for those datasets was less reliable.

In Fig. 2, we show examples of the anomaly score attributions. The right panel of the figure reports the normalized absolute values of the attributions for the *GMM* detector on the `Thyroid` dataset. Each row corresponds to each of four data points, which are emphasized in the scatter plot matrix in the left panel by the associated markers (●, ■, ▲, and ✖). Here, we note that all of these four points are anomalous data points; attributing anomaly scores only makes sense for anomalous queries. In the first two examples (● and ■), the largest attribution by ASH successfully coincides with that of the supervised classifier. In contrast, they do not match in the last two examples (▲ and ✖). We would note the following observations:

- The Shapley-value-based attributions (i.e., IG, KSH, wKSH, and ASH) tend to give relatively (close to) sparse attributions compared to the other methods. This tendency was commonly observed in other examples not shown here, too.

- The attributions by KSH for the first two examples, ● and ■, are similar. In contrast, the attributions by wKSH and ASH are very different between these examples. Such a difference of the behavior can be explained by the fact that while KSH uses the fixed reference value in computing the characteristic function, the reference values used by wKSH and ASH are adaptive to the query data point.

- In the third example (▲), the attribution by ASH significantly deviates from the supervised classifier's attribution. This may be due to the extreme value of the corresponding data point, the point ▲ in the scatter plot matrix. At the same time, the ASH's attribution (largest for Feature #3) is not necessarily meaningless because the point ▲ exhibits an extreme value also for Feature #3.

- In contrast to the third example (▲), the data point of the fourth example (✖) has no clear extreme values and rather lies in the proximity of points ● and ■. This fact makes the reason why ASH failed with the point ✖ less clear. Meanwhile, the failure is lighter than in the previous case because Feature #0 has the second largest value in the attribution by ASH.

### 5.4 Validity of heuristic relaxation

Recall that in Section 3.1, we introduced a heuristic relaxation of the characteristic function. We investigated the validity of the relaxation by comparing the original relaxed characteristic function (i.e., $\hat{v}$ in Eq. (3)) and the heuristically approximated one (i.e., $\underline{v}$ in Eq. (5)). In Table 5, we compare the Shapley value-based attribution using $\hat{v}$ and $\underline{v}$. In terms of the computation time for each $\boldsymbol{x}$, the attribution with $\underline{v}$ is significantly faster than that with $\hat{v}$, which is a natural consequence of the definitions. Meanwhile, the resulting MRRs with each characteristic function (in the setting of synthetic anomaly experiment in Section 5.2) are similar to each other. We report the comparison only for the two datasets in Table 5 because the computation of $\hat{v}$ for datasets with more features was basically infeasible.

## 6 Discussion

The proposed strategy, ASH, resulted in a relatively stably good performance. Nonetheless, it sometimes failed while others were good (though the same could be said for all the methods; no one wins always). These observations tell us that using multiple attribution methods with multiple detection methods in ensembles may be good practice for interpreting anomaly detection.

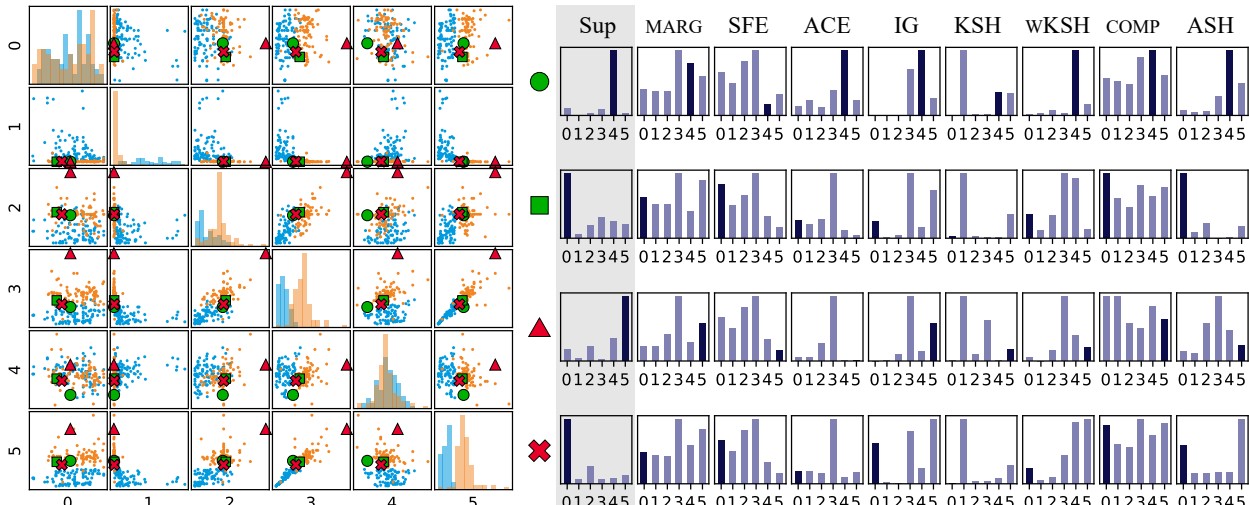

Figure 2: (*Left*) Scatter plot matrix of the `Thyroid` dataset ($d = 6$). The blue and orange points are from $\mathcal{D}_{\text{test}}^{\text{norm}}$ and $\mathcal{D}_{\text{test}}^{\text{anom}}$, respectively. The labels from 0 to 5 are the feature index. (*Right*) Examples of the anomaly score attributions for the *GMM* detector. The normalized absolute values of the attributions are reported. "Sup" refers to the reference attribution of the supervised classifier, $\{\phi_i^{\text{sup}}\}$. Each row corresponds to a data point of the `Thyroid` dataset. The associated markers (●, ■, ▲, and ✖) point to the location of the corresponding data points in the left panel. Note that the points with those markers are anomalous data.

Table 5: Comparison of the two definitions of the characteristic functions for anomaly scores, $\hat{v}$ (without the heuristics, in Eq. (3)) and $\underline{v}$ (with the heuristics, in Eq. (5)). The per-sample computation time and the performance of the attribution for the synthetic anomaly with $d_{\text{anom}} = 1$ are reported.

| Setting | | Average computation time (STD) [sec/sample] | | MRR | |
|---|---|---|---|---|---|
| Dataset | Detector | with $\hat{v}$ | with $\underline{v}$ | with $\hat{v}$ | with $\underline{v}$ |
| Thyroid $d = 6$ | *GMM* | 26.40 (13.19) | 6.19 (3.05) | 0.76 | 0.78 |
| | *VAE-r* | 23.50 (12.06) | 5.58 (3.04) | 0.75 | 0.75 |
| | *VAE-e* | 23.81 (13.50) | 5.42 (3.28) | 0.75 | 0.75 |
| | *DAGMM* | 51.52 (37.19) | 12.45 (8.43) | 0.57 | 0.51 |
| Breastw $d = 9$ | *GMM* | 278.84 (169.07) | 7.10 (4.28) | 0.78 | 0.78 |
| | *VAE-r* | 397.87 (235.73) | 14.76 (8.21) | 0.81 | 0.77 |
| | *VAE-e* | 478.76 (272.03) | 15.86 (8.94) | 0.84 | 0.85 |
| | *DAGMM* | 1071.99 (596.88) | 18.37 (12.48) | 0.57 | 0.57 |

Analysis of each method's failure is an important question, though it is out of the scope of this paper because such failure analysis is meaningful only with in-depth experimentation for each particular anomaly detector, characteristic function, data, and anomaly type. This paper, instead, has provided a general overview of the relevant attribution methods to investigate their applicability. Although we provided examples of attributions in Fig. 2, we think it is not safe to say anything more detailed than what we listed in Section 5.3 because the visualization reveals only limited aspects of the dataset, and the true cause of anomalies cannot necessarily be ascribed to a single feature as we did.

In the experiment, we also observed that the simplest approach, MARG, worked well in some cases. It is, in a sense, reassuring because MARG has been the only way of anomaly attribution practiced in many use cases for a long time. However, it was sometimes substantially outperformed by other methods and can only be computed for limited types of anomaly scores, which motivates the use of the other attribution methods including the proposed one.

Another general observation is that the anomaly score by *DAGMM* was relatively difficult to attribute, especially when $d > 10$, probably because of its strong non-additive nature. Such "attribution hardness" would be an interesting topic of future studies.

Finally, as stated earlier, the evaluation using the real anomaly data has inherent limitations (as is often the case with the evaluation of interpretation methods), since the ground truth was a surrogate. More specific evaluations with real anomalies should be done in each application domain with experts' supervision.

**Broader Impact Statement**

This paper presents some methods for interpreting anomaly detection results, which can be utilized in various real-world domains including industrial sectors. The users must be constantly aware of the heuristic and data-driven nature of the methods; the correctness of the interpretation can never be guaranteed automatically. The methods can be useful in helping the user's decision-making processes but cannot replace any critical roles of human operators. Moreover, as advised in the paper, multiple attribution methods should be used together in ensembles for gaining stability.

**Acknowledgments**

The major part of this work was conducted when the first author was working at RIKEN Center for Advanced Intelligence Project. Afterward, major revision and additional experiments were done while the first author was at the University of Applied Sciences Western Switzerland.

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

## A    Experimental Settings

### A.1    Datasets

`Thyroid` is originally from the UCI machine learning repository (Dua & Graff, 2017), and we used the reformatted dataset provided as a part of the ODDS repository (Rayana, 2016). The reformatted dataset comprises six real-valued features, eliminating the 15 categorical features of the original. The original task for which the dataset was prepared is to determine whether a patient is hypothyroid or not. For anomaly detection purposes, within the three classes of normal functioning, subnormal functioning, and hyperfunction, the first two are used as normal data, and the last is as anomalous data.

`BreastW` is originally from the UCI machine learning repository, and we used the reformatted dataset provided as a part of the ODDS repository. The dataset comprises nine features that take categorical values from 1 to 10. The original problem is the classification between benign and malignant classes. For anomaly detection purposes, the malignant class is used as anomalous data.

`U2R` is a part of the NSL-KDD dataset (Tavallaee et al., 2009), which is a modified version of the KDD Cup 1999 dataset. From the NSL-KDD dataset, we used the part of the dataset corresponding to the U2R attack type. We eliminated the six categorical features and a real-valued feature that does not change in the dataset, which resulted in the following ten features (in the original names):

1. *duration*
2. *hot*
3. *num_compromised*
4. *root_shell*
5. *num_root*
6. *num_file_creations*
7. *srv_count*
8. *dst_host_count*
9. *dst_host_srv_count*
10. *dst_host_same_src_port_rate*

`Lympho` is originally from the UCI machine learning repository, and we used the reformatted dataset provided as a part of the ODDS repository. The dataset comprises 18 categorical features, and we transformed them by one-hot encoding, which resulted in the 59-dimensional dataset. The original task is the classification between four classes, two of which are quite small. For anomaly detection purposes, these small classes are used as anomalous data.

`Musk` is originally from the UCI machine learning repository, and we used the reformatted dataset provided as a part of the ODDS repository. The dataset comprises 166 real-valued features. The original task is to

classify molecules into musk and non-musk classes. For anomaly detection purposes, three non-musk classes are used as normal data, and two musk classes are as anomalous data.

`Arrhythmia` is originally from the UCI machine learning repository, and we used the reformatted dataset provided as a part of the ODDS repository. The reformatted dataset comprises 274 real-valued features. The original task is a 16-class classification to distinguish between the presence and absence of cardiac arrhythmia. For anomaly detection purposes, the eight smallest classes are used as anomalous data.

### A.2 Anomaly detection methods

#### A.2.1 Gaussian mixture models (GMMs)

We selected best numbers of mixture components, $K$, from 2, 3, or 4, based on the validation likelihood. The corresponding anomaly score is the energy (i.e., negative log-likelihood) of the GMMs:

$$e(\boldsymbol{x}) = -\log \sum_{k=1}^{K} \pi_k \exp\left( -\frac{d\log(2\pi)}{2} - \frac{1}{2}\log\det(\boldsymbol{\Sigma}_k) - \frac{1}{2}(\boldsymbol{x} - \boldsymbol{\mu}_k)^\mathsf{T} \boldsymbol{\Sigma}_k^{-1}(\boldsymbol{x} - \boldsymbol{\mu}_k) \right),$$

where $\{\pi_1, \ldots, \pi_K\}$ is the set of mixture weights, $\{\boldsymbol{\mu}_1, \ldots, \boldsymbol{\mu}_K\}$ is the set of means, and $\{\boldsymbol{\Sigma}_1, \ldots, \boldsymbol{\Sigma}_K\}$ is the set of covariance matrices.

#### A.2.2 Variational autoencoders (VAEs)

The encoder is a multilayer perceptron with one hidden layer, whereas the decoder is with two hidden layers. Every activation function is the softplus function. We selected the best values of the dimensionality of the latent variable, $\dim(\boldsymbol{z})$, and the number of the hidden units of the multilayer perceptrons, $\dim(\text{MLP})$, based on the validation ELBO. The candidate of $\dim(\boldsymbol{z})$ was the rounded values of $0.2d$, $0.4d$, $0.6d$, and $0.8d$, where $d$ is the dimensionality of each dataset. The candidate of $\dim(\text{MLP})$ was the rounded values of $0.5d$, $d$, and $2d$. The loss function used for learning was the negative ELBO with the mean squared loss for the real-valued datasets or with the cross-entropy loss for the binary-valued dataset. We used the Adam optimizer with the learning rate 0.001 and stopped the optimization observing the validation set loss. The corresponding anomaly scores are:

$$e(\boldsymbol{x}) = \left\| \boldsymbol{x} - \boldsymbol{f}(\boldsymbol{g}(\boldsymbol{x})) \right\|^2$$

and

$$e(\boldsymbol{x}) = -\mathbb{E}_{q_\psi(\boldsymbol{z}|\boldsymbol{x})}\big[ \log p_\theta(\boldsymbol{x} \mid \boldsymbol{z}) + \log p(\boldsymbol{z}) - \log q_\psi(\boldsymbol{z} \mid \boldsymbol{x}) \big],$$

for *VAE-r* and *VAE-e*, respectively. $\theta$ and $\psi$ denote the sets of parameters of the decoder and the encoder, respectively. $\boldsymbol{g}$ and $\boldsymbol{f}$ are the decoder and the encoder's mean parameter function, respectively.

#### A.2.3 Deep autoencoding Gaussian mixture models (DAGMMs)

The architectures of the encoder and decoder are the same as the VAEs above. What is specific to DAGMMs is the so-called estimation network that outputs the estimation of cluster assignment of a GMM learned on the latent representations and the reconstruction error values. In our experiments, the estimation network is a multilayer perceptron with one hidden layer using the softplus function as activation. The candidates of the hyperparameters were the same as in the above cases, both for the autoencoder part and the GMM part. We used the Adam optimizer with the learning rate 0.0001 and stopped the optimization observing the validation set loss.

## B Experimental Results

### B.1 Sensitivity to hyperparameter

In Table 6, we report results of the synthetic anomaly experiment (in Section 5.2) using the *GMM* detector with $\gamma$ being varied from 0.001 to 10. We can observe that the performance is insensitive to $\gamma$.

Table 6: MRR for the synthetic anomaly. The values for the *GMM* detector using different values of $\gamma$ are reported.

| Dataset | MRR | | | | |
| --- | --- | --- | --- | --- | --- |
| | $\gamma = 0.001$ | $\gamma = 0.01$ | $\gamma = 0.1$ | $\gamma = 1.0$ | $\gamma = 10.0$ |
| Thyroid | 0.77 | 0.78 | 0.78 | 0.75 | 0.74 |
| Breastw | 0.78 | 0.78 | 0.78 | 0.77 | 0.78 |
| U2R | 0.86 | 0.86 | 0.86 | 0.86 | 0.86 |
| Lympho | 0.73 | 0.73 | 0.73 | 0.73 | 0.74 |
| Musk | 0.98 | 0.97 | 0.95 | 0.93 | 0.92 |
| Arrhythmia | 0.49 | 0.49 | 0.49 | 0.50 | 0.50 |

Table 7: Per-sample computation time of the attribution with the proposed characteristic function, $\underline{v}$.

| Setting | | Average computation time (STD) |
| --- | --- | --- |
| Dataset | Detector | [sec/sample] |
| Thyroid $d = 6$ | *GMM* | 6.19 (3.05) |
| | *VAE-r* | 5.58 (3.04) |
| | *VAE-e* | 5.42 (3.28) |
| | *DAGMM* | 12.45 (8.43) |
| Breastw $d = 9$ | *GMM* | 7.10 (4.28) |
| | *VAE-r* | 14.76 (8.21) |
| | *VAE-e* | 15.86 (8.94) |
| | *DAGMM* | 18.37 (12.48) |
| U2R $d = 10$ | *GMM* | 10.35 (7.64) |
| | *VAE-r* | 15.13 (7.30) |
| | *VAE-e* | 11.53 (6.36) |
| | *DAGMM* | 34.69 (18.01) |
| Lympho $d = 59$ | *GMM* | 81.41 (33.14) |
| | *VAE-r* | 213.68 (0.17) |
| | *VAE-e* | 226.29 (0.27) |
| | *DAGMM* | 96.68 (32.34) |
| Musk $d = 166$ | *GMM* | 217.82 (103.43) |
| | *VAE-r* | 184.01 (62.34) |
| | *VAE-e* | 464.39 (92.27) |
| | *DAGMM* | 503.65 (85.80) |
| Arrhythmia $d = 274$ | *GMM* | 1187.44 (418.56) |
| | *VAE-r* | 580.67 (298.37) |
| | *VAE-e* | 1480.86 (161.63) |
| | *DAGMM* | 404.46 (270.18) |

## B.2 Runtime

With the heuristic relaxation we introduced (i.e., from Eq. (3) to Eq. (4)), the number of the minimization problems to be solved decreases from $O(2^d)$ to $O(d)$. While the overall time complexity of the attribution algorithm certainly depends on this number, the actual runtime depends also on many other factors. Specifically, each of the minimization problems is solved with gradient descent, whose stopping rule refers to the empirical convergence of the objective, and thus the number of iterations significantly differs with different data. We report the average computation time of the attribution method with the proposed characteristic function (i.e., ASH) for the six datasets. The overall averages of the computation time over the four detectors are summarized in Fig. 3. Note that the evaluation for the last two datasets ($d = 166$ and $d = 274$) is difficult because they show large variances.

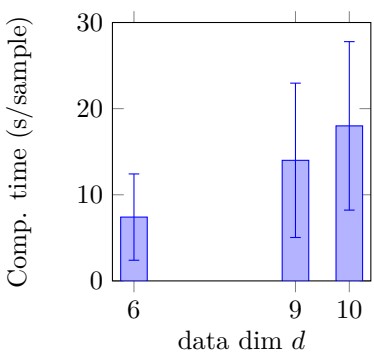 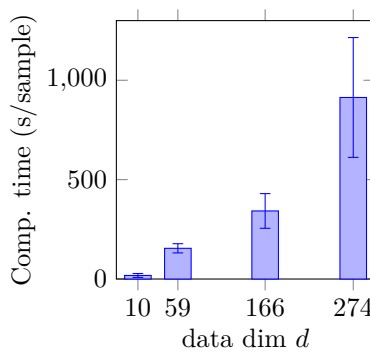

Figure 3: Per-sample computation time reported in Table 7. The overall averages over the four detectors are plotted. The left panel is for the first three datasets, and the right panel is for the last four datasets.

Table 8: Average AUROC for synthetic anomaly with $d_{\mathrm{anom}} = 2$.

| Setting | | AUC | | | | | | | |
|---|---|---|---|---|---|---|---|---|---|
| Dataset | Detector | MARG | SFE | ACE | IG | KSH | wKSH | COMP | ASH |
| Thyroid | GMM | 0.73 | 0.73 | 0.78 | 0.76 | 0.56 | 0.73 | 0.52 | 0.83 |
| | VAE-r | 0.82 | 0.82 | 0.83 | 0.82 | 0.68 | 0.80 | 0.82 | 0.82 |
| | VAE-e | — | — | 0.82 | 0.82 | 0.67 | 0.80 | 0.48 | 0.82 |
| | DAGMM | — | — | 0.43 | 0.17 | 0.49 | 0.54 | 0.73 | 0.58 |
| Breastw | GMM | 0.90 | 0.89 | 0.85 | 0.89 | 0.60 | 0.86 | 0.48 | 0.89 |
| | VAE-r | 0.77 | 0.77 | 0.70 | 0.87 | 0.81 | 0.86 | 0.88 | 0.83 |
| | VAE-e | — | — | 0.68 | 0.89 | 0.81 | 0.85 | 0.51 | 0.86 |
| | DAGMM | — | — | 0.48 | 0.13 | 0.58 | 0.73 | 0.53 | 0.68 |
| U2R | GMM | 0.95 | 0.88 | 0.68 | 0.90 | 0.53 | 0.86 | 0.54 | 0.89 |
| | VAE-r | 0.91 | 0.91 | 0.89 | 0.92 | 0.59 | 0.89 | 0.94 | 0.93 |
| | VAE-e | — | — | 0.87 | 0.93 | 0.60 | 0.88 | 0.46 | 0.92 |
| | DAGMM | — | — | 0.59 | 0.06 | 0.50 | 0.71 | 0.92 | 0.73 |
| Lympho | GMM | 0.89 | 0.93 | 0.70 | 0.97 | 0.50 | 0.50 | 0.63 | 0.95 |
| | VAE-r | 0.99 | 0.99 | 0.71 | 0.90 | 0.50 | 0.50 | 0.29 | 0.95 |
| | VAE-e | — | — | 0.71 | 0.93 | 0.50 | 0.50 | 0.29 | 0.93 |
| | DAGMM | — | — | 0.60 | 0.40 | 0.50 | 0.50 | 0.71 | 0.86 |
| Musk | GMM | 0.69 | 0.69 | 0.65 | 0.81 | 0.74 | 0.95 | 0.48 | 0.96 |
| | VAE-r | 0.99 | 0.99 | 0.92 | 0.95 | 0.90 | 0.99 | 0.75 | 1.00 |
| | VAE-e | — | — | 0.91 | 0.95 | 0.90 | 0.99 | 0.48 | 0.99 |
| | DAGMM | — | — | 0.64 | 0.25 | 0.55 | 0.75 | 0.48 | 0.77 |
| Arrhythmia | GMM | 0.90 | 0.87 | 0.86 | 0.90 | 0.73 | 0.90 | 0.48 | 0.91 |
| | VAE-r | 0.98 | 0.98 | 0.92 | 0.94 | 0.89 | 0.93 | 0.85 | 0.96 |
| | VAE-e | — | — | 0.92 | 0.94 | 0.89 | 0.93 | 0.52 | 0.95 |
| | DAGMM | — | — | 0.57 | 0.11 | 0.62 | 0.67 | 0.48 | 0.68 |
| 25th percentile | | (0.81) | (0.81) | 0.65 | 0.67 | 0.52 | 0.70 | 0.48 | **0.82** |
| 50th percentile | | (0.90) | (0.89) | 0.71 | **0.89** | 0.60 | 0.82 | 0.52 | **0.89** |
| 75th percentile | | (0.96) | (0.94) | 0.86 | 0.93 | 0.76 | 0.89 | 0.73 | **0.95** |

## B.3 Other performance measures

We show the results of the synthetic anomaly experiment (in Section 5.2) with $d_{\mathrm{anom}} > 1$. Since the MRR and the Hits@3 are not necessarily meaningful when $d_{\mathrm{anom}} > 1$, we report the area under the receiver operator characteristic curve (AUROC) in Tables 8 and 9; given a set of attributions to features, we sweep a threshold value for the attributions from the largest attribution to the smallest attribution to define the ROCs. The overall tendency of the performance is the same as we reported in Section 5.2.

Table 9: Average AUROC for synthetic anomaly with $d_{\mathrm{anom}} = 3$.

| Setting | | AUC | | | | | | | |
|---|---|---|---|---|---|---|---|---|---|
| Dataset | Detector | MARG | SFE | ACE | IG | KSH | wKSH | COMP | ASH |
| Thyroid | GMM | 0.74 | 0.71 | 0.77 | 0.75 | 0.58 | 0.71 | 0.50 | 0.82 |
| | VAE-r | 0.85 | 0.85 | 0.82 | 0.85 | 0.66 | 0.81 | 0.85 | 0.85 |
| | VAE-e | — | — | 0.82 | 0.85 | 0.66 | 0.80 | 0.46 | 0.85 |
| | DAGMM | — | — | 0.52 | 0.16 | 0.47 | 0.57 | 0.75 | 0.67 |
| Breastw | GMM | 0.91 | 0.89 | 0.85 | 0.90 | 0.63 | 0.85 | 0.46 | 0.91 |
| | VAE-r | 0.74 | 0.74 | 0.63 | 0.81 | 0.79 | 0.79 | 0.91 | 0.81 |
| | VAE-e | — | — | 0.63 | 0.84 | 0.79 | 0.80 | 0.53 | 0.81 |
| | DAGMM | — | — | 0.53 | 0.10 | 0.60 | 0.69 | 0.53 | 0.66 |
| U2R | GMM | 0.96 | 0.82 | 0.64 | 0.87 | 0.53 | 0.84 | 0.48 | 0.83 |
| | VAE-r | 0.92 | 0.92 | 0.89 | 0.94 | 0.57 | 0.92 | 0.94 | 0.94 |
| | VAE-e | — | — | 0.85 | 0.95 | 0.57 | 0.91 | 0.52 | 0.94 |
| | DAGMM | — | — | 0.56 | 0.06 | 0.52 | 0.66 | 0.92 | 0.68 |
| Lympho | GMM | 0.89 | 0.91 | 0.71 | 0.96 | 0.50 | 0.50 | 0.58 | 0.93 |
| | VAE-r | 0.97 | 0.97 | 0.71 | 0.88 | 0.50 | 0.50 | 0.29 | 0.91 |
| | VAE-e | — | — | 0.71 | 0.91 | 0.50 | 0.50 | 0.29 | 0.88 |
| | DAGMM | — | — | 0.66 | 0.44 | 0.50 | 0.50 | 0.71 | 0.82 |
| Musk | GMM | 0.68 | 0.71 | 0.71 | 0.83 | 0.73 | 0.96 | 0.47 | 0.95 |
| | VAE-r | 1.00 | 1.00 | 0.91 | 0.95 | 0.90 | 1.00 | 0.75 | 0.99 |
| | VAE-e | — | — | 0.91 | 0.95 | 0.90 | 1.00 | 0.47 | 0.99 |
| | DAGMM | — | — | 0.70 | 0.25 | 0.58 | 0.77 | 0.47 | 0.79 |
| Arrhythmia | GMM | 0.91 | 0.88 | 0.87 | 0.91 | 0.71 | 0.92 | 0.47 | 0.91 |
| | VAE-r | 0.98 | 0.98 | 0.91 | 0.94 | 0.88 | 0.95 | 0.87 | 0.98 |
| | VAE-e | — | — | 0.91 | 0.94 | 0.88 | 0.95 | 0.53 | 0.97 |
| | DAGMM | — | — | 0.56 | 0.10 | 0.61 | 0.68 | 0.47 | 0.69 |
| 25th percentile | | (0.82) | (0.80) | 0.64 | 0.67 | 0.53 | 0.68 | 0.47 | **0.81** |
| 50th percentile | | (0.91) | (0.89) | 0.71 | **0.86** | 0.60 | 0.80 | 0.53 | **0.86** |
| 75th percentile | | (0.96) | (0.93) | 0.85 | **0.94** | 0.74 | 0.92 | 0.75 | **0.94** |

