# OpenReview forum: "A Characteristic Function for Shapley-Value-Based Attribution of Anomaly Scores"
_TMLR — Accepted by TMLR_

### Review · Reviewer_4agy · 2023-02-28

**Summary Of Contributions:**

This paper proposes a new framework for the task of model agnostic (black box) anomaly attribution based on Shapley value (SV). Unlike prior methods that simply use an existing implementation or formula ironically as a black box, the authors look deep into the very definition of SV. In particular, they propose a new definition of SV’s characteristic function that depends on the local minimum of the anomaly score. More specifically, the key ideas are:

1. To use the SV value for the anomaly scoring function rather than the prediction function itself,
2. To use a characteristic function that depends on a “reference vector”, and
3. To determine the reference vector as a local minimum point of the anomaly score.

Although they propose a well-defined optimization problem in Eq. (3) as the canonical definition of the proposed framework, it is not something they actually used in the empirical evaluation. To handle computational difficulties, the authors introduce two heuristic approaches. Instead of solving the regularized optimization problem (3), they propose to use a combination between the all-variable local minimum and one-variable conditional minima. Also, they use a SHAP-like least squared variant of SV.

**Audience:**

Yes

**Broader Impact Concerns:**

Except for very few exceptions, almost all model-agnostic anomaly explanation papers for black-box models ironically use existing attribution formulas/codes as a black box. This paper shines in such unfortunate darkness.

**Claims And Evidence:**

Yes

**Requested Changes:**

This paper can be accepted as-is.

This is not a mandatory request at all, but I encourage the authors to address the unclarity listed above. It will improve the readability of the paper significantly.

**Strengths And Weaknesses:**

**Strength**

The idea of

- applying SV to the anomaly score rather than the prediction function and
- using the local minimum as the reference point to define the characteristic function of SV

is quite unique. I found the novelty and the originality of the paper very high. The idea alone definitely deserves immediate acceptance to top ML journals like TMLR.


**Weakness**

However, although this work indeed has the potential to become the first crucial work on SV-based attribution since SV's introduction to the ML community, I’d point out that the treatment after Eq. (3) looks rough. My biggest complaint is that the authors do not show the relationship between (3) and (4). I don’t even understand if it is an approximation or something else. More specifically,

- I couldn’t find any clear description of the choice of dist().
- I couldn’t find any clear description of the choice of $\gamma$.
- The second condition in (4) is not even the average, as the summation runs over $2S$ terms. A clear justification is needed.
- No clear justification is provided about mixing the two local minima between those for $\{i\}$ and $\emptyset$.
- No clear justification is given for the use of the least square “approximation” of SV. I’m not sure if Lundberg’s paper uses it as an approximation. This is not the authors’ fault, but Lundberg’s paper does not provide a clear-cut definition of the characteristic function, and its discussion is logically hard to follow. I do not think it is a good idea to use it blindly without justification.

---

> ### Author Response · Authors · 2023-05-01
> **Response to Reviewer 4agy**
>
> Thank you so much for reviewing our paper.
> We appreciate the very favorable comments.
> We addressed each of the reviewer's comments.
> We emphasized the corresponding parts of the revised paper in blue or magenta.
>
> > ... the treatment after Eq. (3) looks rough.
> > My biggest complaint is that the authors do not show the relationship between (3) and (4).
> > I don't even understand if it is an approximation or something else.
>
> We agree that our heuristic relaxation of the formulation from Eq. (3) to Eq. (4) lacks a very good theoretical justification.
> We did not intend to claim that Eq. (4) is an approximation of Eq. (3) in some proper sense. However, we still said "approximation" mistakenly in some parts; we reworded them by "relaxation" in Section 5.4 (Page 12; emphasized in magenta).
>
> We empirically found that the heuristic relaxation still worked nicely to some extent, which is a part of the main contribution of the paper.
> With that being said, the original formulation in Eq. (3) seems more natural, and a more rigorous method to make it computationally feasible is to be addressed.
>
> > I couldn't find any clear description of the choice of dist().
>
> The dissimilarity measure, $\operatorname{dist}()$, is to be chosen in accordance with the nature of the data space.
> If data are embedded in $\mathbb{R}^d$, the natural choice would be the Euclidean distance.
> We revised the description to make it clear in Section 3.1, the paragraph below Eq. (3) (Page 5; emphasized in magenta).
>
> > I couldn't find any clear description of the choice of $\gamma$.
>
> Selecting the optimal value of $\gamma$ automatically is challenging because it depends on the geometry of the data space.
> A practical guideline is to create anomalous data artificially based on prior knowledge, try different values of $\gamma$, and see how well the artificial anomalies are attributed.
> We added a description in this regard in Section 3.1, the paragraph below Eq. (3) (Page 5; emphasized in blue).
>
> >  The second condition in (4) is not even the average, as the summation runs over $2S$ terms. A clear justification is needed.
>
> Sorry, the confusion is due to a typo. The place of the parenthesis was wrong in the latter part of Eq. (4).
> The summation runs over $|S|+1$ terms because it is over the cases with the first argument of $\hat{\boldsymbol{x}}^\star$ being the empty set (1 case) or the singleton of each element of $S$ ($|S|$ cases).
> We corrected Eq. (4) accordingly (Page 5; emphasized in magenta).
>
> > No clear justification is provided about mixing the two local minima between those for $i$ and $\emptyset$.
>
> In Eq. (4), the relaxed definition mixes not only two but $|S|+1$ local solutions of the minimization of $\ell_{\cdot,\boldsymbol{x}}$.
> The idea here is as follows.
> First, recall that the original definition in Eq. (3) is the solution of the minimization problem with **all** the elements of $\boldsymbol{x}$ indexed in $S$ being fixed.
> Then, in Eq. (4), we make its surrogate by averaging the local solutions of the problems with **each** element of $\boldsymbol{x}$ in S being fixed.
> Here we counted the empty set as one of the cases of this averaging process, thus $|S|+1$ cases.
> With that being said, this is nothing more than heuristics as mentioned earlier, and not including the empty set as a case might work well, too.
> We added a paragraph for this explanation at the end of Section 3.1 (Page 5; emphasized in blue).
>
> > No clear justification is given for the use of the least square "approximation" of SV. I'm not sure if Lundberg's paper uses it as an approximation. This is not the authors' fault, but Lundberg's paper does not provide a clear-cut definition of the characteristic function, and its discussion is logically hard to follow. I do not think it is a good idea to use it blindly without justification.
>
> Thank you for pointing this out.
> We agree we should not say "approximation as least squares" because it mixes up two different things without a clear explanation.
> In the paper (Lundberg & Lee 2017), in their Theorem 2, they show that the weighted least squares formulation is *equivalent to* the Shapley value's original definition.
> We also adopted this formulation for convenience in coding, but before solving the weighted least squares problem, we randomly draw subsets of $S$.
> Thus, we should say that *we use the least squares formulation upon the Monte Carlo approximation*.
> We modified the description in Section 3.2 accordingly (Page 6; emphasize in blue).
>
> By the way, the above argument holds regardless of the definition of the characteristic function, $v$, because the way we subsample the subsets of $S$ is agnostic of any properties of $v$.
> For the definition of $v$, we proposed a way specifically for attributing anomaly scores, instead of adopting the general approaches discussed in Lundberg & Lee (2017) and other literature.

---

> > ### Comment · Reviewer_4agy · 2023-05-01
> > **I'm good now. Thank you.**
> >
> > Thank you for your detailed explanation. I may ask you about some details later, but I feel like you have addressed all the points I raised. I will endorse acceptance of this paper in the next discussion stage for sure.

---

### Review · Reviewer_6xJf · 2023-03-12

**Summary Of Contributions:**

The authors consider the problem of attribution in anomaly detection. The objective is to identify which features lead to a particular sample being detected as anomalous. They propose to perform this attribution using Shapley values, which are frequently used for attribution in supervised learning tasks. To adapt the approach to anomaly detection, they propose a new characteristic function along with a faster heuristic relaxation for it. They demonstrate that their proposed anomaly Shapley (ASH) attribution method performs well for attributing synthetically generating anomalies by perturbing features on real data.

The main contributions I observe are as follows:
- Proposal of a new reference-based characteristic function for anomaly scores to be used for computing Shapley values.
- Proposal of a heuristic relaxation for the characteristic function that is computationally efficient.
- Demonstration of effectiveness of proposed ASH relaxed characteristic function on real data sets using two different evaluation approaches.

*After author revision:* I have changed Claims and Evidence from No to Yes given the newly added examples and experiment results. My concerns regarding the paper have been addressed.

**Audience:**

Yes

**Broader Impact Concerns:**

The included statement is reasonable.

**Claims And Evidence:**

Yes

**Requested Changes:**

Major issues:
- I strongly recommend for the authors to add a case study showing Shapley values computed on one of the real datasets. Perhaps this could be done in the same setting as in Section 5.3, where they compare the attribution performance with that of a supervised classifier. It would be highly useful to see the distributions of Shapley values with the different attribution methods to see how they differ and perhaps why ASH performs well. This can be added without removing content from the current paper and staying within 12 pages of main body.
- The authors state that analysis of each method's failure is out of the scope of this paper. I agree that a detailed analysis of the type they describe is out of scope; however, an illustration of instances where different attribution methods differ, perhaps as part of the case study I suggest above, would be useful and very informative.

Minor issues:
- An experiment validating the $O(d)$ time complexity of the heuristic relaxation compared to the $O(2^d)$ complexity without the relaxation would be useful as an addition to the appendix. We can get a limited idea of the empirical time complexity from Table 5, but it's only 2 data points for $d$.
- In Table 1, it would be good to show also a column with the size of $\mathcal{D}_{\text{test}}^{\text{anom}}$ so that the reader can see how balanced the nominal and anomalous classes are.
- Why do you only extract U2R attack type from the NSL-KDD data rather than using all of the different attacks?


**Strengths And Weaknesses:**

Strengths:
- The authors design a characteristic function for computing Shapley values that is specifically target at anomaly attribution.
- Strong empirical performance in 2 different settings. It is not possible to compare attribution accuracy on real anomaly data because there is no ground truth to compare against, so the authors use both synthetically generated anomalies (where they can control the anomalous features) and real anomalies while treating the attributions from a supervised classifier as the "ground truth" for comparison.

Weaknesses:
- The paper does not show the computed Shapley values for any datasets. This is very strange given that the main purpose of this paper is about attribution, but then it doesn't show any attribution on real data.
- No experimental validation of time complexity for the proposed heuristic relaxation is provided.

---

> ### Author Response · Authors · 2023-05-01
> **Response to Reviewer 6xJf**
>
> Thank you so much for reviewing our paper.
> We appreciate the constructive feedback.
> We addressed each of the reviewer's comments.
> We emphasized the corresponding parts of the revised paper in blue.
>
> > I strongly recommend for the authors to add a case study showing Shapley values computed on one of the real datasets. Perhaps this could be done in the same setting as in Section 5.3, where they compare the attribution performance with that of a supervised classifier. It would be highly useful to see the distributions of Shapley values with the different attribution methods to see how they differ and perhaps why ASH performs well. This can be added without removing content from the current paper and staying within 12 pages of main body.
>
> Thank you for the great suggestion.
> We added examples of the attributions in the new Fig. 2.
> They are for 4 different data points of the Thyroid dataset, which is relatively easy to visualize.
> For the details, please refer to Fig. 2 and the last part of Section 5.3 of the revised paper (Page 11).
>
> > The authors state that analysis of each method's failure is out of the scope of this paper. I agree that a detailed analysis of the type they describe is out of scope; however, an illustration of instances where different attribution methods differ, perhaps as part of the case study I suggest above, would be useful and very informative.
>
> The new Fig. 2 includes the cases where the proposed method, ASH, failed to identify the feature that the supervised classifier did.
> We can observe that a severe failure (the 3rd example, triangle) happened for a data point lying far from the dense region of the data.
> A failure happened also for a data point in a relatively dense region (the 4th example, cross mark), but in this case, we can say that the extent of the failure was milder because the feature to be most attributed has the second largest attribution.
> This observation implies that attributing anomaly scores by the proposed method may tend to be difficult when the query data is extreme.
> We added this observation in the list in the last paragraph of Section 5.3 (Page 11).
>
> > An experiment validating the $O(d)$ time complexity of the heuristic relaxation compared to the $O(2^d)$ complexity without the relaxation would be useful as an addition to the appendix. We can get a limited idea of the empirical time complexity from Table 5, but it's only 2 data points for $d$.
>
> Thank you for the suggestion.
> With the heuristic relaxation, *the number of the minimization problems to be solved* decreases from $O(2^d)$ to $O(d)$.
> Although the overall time complexity of the attribution algorithm depends on this number, empirically validating such a dependency is not straightforward because the actual runtime depends also on many other factors.
> Specifically in our case, each of the minimization problems is solved with gradient descent, whose stopping rule refers to the empirical convergence of the objective, and thus the number of iterations significantly differs with different data.
>
> With that being said, we agree that more information on the runtime may be useful.
> We added a table reporting the runtime for all the datasets and a figure that summarizes the runtime against the data dimensionality $d$ in the new Section B.2 in the appendix (Pages 16-18).
> It basically looks to increase roughly linearly (or slightly faster, probably due to some overhead in our suboptimal implementation) and certainly slower than the exponential increase.
> Please note that empirically assessing the complexity is still difficult due to the large variance of the runtime with larger $d$.
>
> > In Table 1, it would be good to show also a column with the size of $\mathcal{D}_\text{test}^\text{anom}$ so that the reader can see how balanced the nominal and anomalous classes are.
>
> As mentioned in the second paragraph of Section 5.1.1, the size of the normal and anomalous portion of the test data is the same because we restructured the original datasets as such.
> So, $\vert \mathcal{D}_\text{test} \vert = 2 \vert \mathcal{D}_\text{test}^\text{norm} \vert = 2 \vert \mathcal{D}_\text{test}^\text{anom} \vert$.
> We made it clear in the caption of Table 1 (Page 7).
>
> > Why do you only extract U2R attack type from the NSL-KDD data rather than using all of the different attacks?
>
> We extracted the U2R attack type because the original NSL-KDD was too large for the purpose of our study.
> It was also possible to extract each small portion of different attack types, but we occasionally did not do so without any strong reason.
> That being said, we believe the current results are more or less sufficient to see the performance of the methods with different types of datasets.

---

> > ### Comment · Reviewer_6xJf · 2023-05-25
> > **No further concerns**
> >
> > Thanks for the detailed rebuttal and revisions. I find the addition of Figure 2 showing a "case study" to be quite well done and useful. I have no further concerns.

---

### Review · Reviewer_mhYD · 2023-04-25

**Summary Of Contributions:**

The paper is concerned with anomaly attribution. Specifically, while the Shapley value has been used for anomaly detection it is difficult to use it for anomaly attribution. The authors present a method for doing that where a minimization problem is solved over the ``absent’’ features. Since the problem in turn is computationally expensive to solve, algorithms to approximate this solution are used and validated empirically.

**Audience:**

Yes

**Broader Impact Concerns:**

I don't think that there are any concerns about the broader impact.

**Claims And Evidence:**

Yes

**Requested Changes:**

A-In the definition of the Shapley value (in Eq(1) page 2), $S$ is not defined at all or mentioned which is confusing.


Minor Points:

B-I think that adding a citation for semi-supervised methods would be good for the presentation. Speicifically in the following sentence in the first paragraph of section 2.1: ``A typical solution of semi-supervised anomaly detection consists of two phases [references to this line of work] ''

C-It seems to me that in the following sentences “and” should be replaced by ‘or’:

1-2nd paragraph in section 2.1:  such as ``$X ⊂ R^d$ and $X ⊂ \{0, 1}^d$ '' → such as $X ⊂ R^d$ or $X ⊂ \{0, 1\}^d$.

2-3rd paragraph in section 2.1: ``such as principal component analysis (PCA) and autoencoders'' → such as principal component analysis (PCA) or autoencoders


D-Section 6 (Discussion): ``$d \ge O(10)$''. Asymptotic notation is meaningless for constants.




**Strengths And Weaknesses:**

A-The idea of minimizing the score function over the other features seems very reasonable and one can say that it is a suitable approach in applications where false positive anomaly detection is costly.

B-My issue is that while the way the optimization in (2) on page 4 is formulated is rigorous, the modifications to (3) and (4) on page 5 feels very heuristic. But it seems that some of the methods here have already been used in the literature such as randomly sampling features in Lundberg et al.

C-I mostly work on the theory side and I’m not that familiar with anamoly detection, so it is not clear to me how significant the contribution of this paper is. It seems to give a performance which is on par or at times better than previous methods. I believe that would be within the TMLR scope.

---

> ### Author Response · Authors · 2023-05-01
> **Response to Reviewer mhYD**
>
> Thank you so much for reviewing our paper.
> We appreciate the reviewer's careful reading.
> We addressed each of the reviewer's comments.
> We emphasized the corresponding parts of the revised paper in blue.
>
> > B-My issue is that while the way the optimization in (2) on page 4 is formulated is rigorous, the modifications to (3) and (4) on page 5 feels very heuristic.
>
> It is right that the modification from Eq. (3) to Eq. (4) is heuristic.
> Our contribution rather lies in empirically validating that such a heuristic modification still works nicely to some extent.
> It is an interesting open problem to make the computation of the original formulation in Eq. (3) feasible in a more rigorous manner.
>
> >A-In the definition of the Shapley value (in Eq(1) page 2), $S$ is not defined at all or mentioned which is confusing.
>
> $S$ is a temporary variable that denotes a subset of $D \backslash${$i$}.
> We made it clear in Section 2.2, just below Eq. (1) (Page 3).
>
> > B-I think that adding a citation for semi-supervised methods would be good for the presentation. Speicifically in the following sentence in the first paragraph of section 2.1: ``A typical solution of semi-supervised anomaly detection consists of two phases [references to this line of work] ''
>
> Thank you for the suggestion.
> We added a reference to the corresponding sections of the well-known classical survey paper (Chandola et al., 2009) in Section 2.1 (Page 2).
> Many methods following the two-phase process appear there.
>
> > C-It seems to me that in the following sentences “and” should be replaced by ‘or’: ...
>
> Thanks for finding this; we accordingly revised the description.
>
> > D-Section 6 (Discussion): ``$d>O(10)$''. Asymptotic notation is meaningless for constants.
>
> It is indeed not a valid notation that came from a rough note.
> In the revised version of Section 6 (Page 12), we now simply say $d>10$ because the sentence intends to summarize the observation that the DAGMM detector was relatively difficult to do the attribution for datasets with $d=59,166,274$ compared to those with $d=6,9,10$.

---

### Decision · Action_Editors · 2023-07-10

**Recommendation:** Accept as is

**Comment:**

This paper studies the problem of attribution in anomaly detection, aiming to identify the important features causing anomaly. The feature attribution is studied via adapting the Shapley values, and a new characteristic function as a faster heuristic relaxation. The approach shows performance gains on attributing synthetically generated anomalies. Three reviewers provided assessment reports for this paper, all favoring the technical contributions while proposing major concerns. These issues were well addressed in the revised version, and all reviewers in general agreed that the paper is accetable. The AE oversaw the reviewing process and all evidences, and supported the Accept decision.

**Audience:**

This paper tackles a real problem of anomaly attribution, that the anomaly detection community will be interested and inspired from this study.

**Claims And Evidence:**

As confirmed by the three reviewers, the claims made in the submission are well supported either theoretically or empirically.